# Investigation into the Hybrid Production of a Superelastic Shape Memory Alloy with Additively Manufactured Structures for Medical Implants

**DOI:** 10.3390/ma14113098

**Published:** 2021-06-05

**Authors:** Isabell Hamann, Felix Gebhardt, Manuel Eisenhut, Peter Koch, Juliane Thielsch, Christin Rotsch, Welf-Guntram Drossel, Christoph-Eckhard Heyde, Mario Leimert

**Affiliations:** 1Fraunhofer Institute for Machine Tools and Forming Technology IWU, 01187 Dresden, Germany; Felix.Gebhardt@iwu.fraunhofer.de (F.G.); manuel-eisenhut@posteo.de (M.E.); juliane.thielsch@iwu.fraunhofer.de (J.T.); christian.rotsch@iwu.fraunhofer.de (C.R.); Welf-Guntram.Drossel@iwu.fraunhofer.de (W.-G.D.); 2Asklepios Orthopädische Klinik Hohwald, 01844 Neustadt in Sachsen, Germany; 3Chair of Engineering Design and CAD, Dresden University of Technology, 01069 Dresden, Germany; peter.koch@tu-dresden.de; 4Professorship Adaptronics and Lightweight Design in Production, Chemnitz University of Technology, 09107 Chemnitz, Germany; 5Medical Center, Orthopaedic, Trauma and Plastic Surgery Clinic, University of Leipzig, 04103 Leipzig, Germany; Christoph-Eckhard.Heyde@medizin.uni-leipzig.de; 6Sächsische Schweiz Kliniken GmbH, 01855 Sebnitz, Germany; m.leimert@asklepios.com

**Keywords:** shape memory alloys (SMA), NiTi, superelasticity, application, medical, implant, additive manufacturing

## Abstract

The demographic change in and the higher incidence of degenerative bone disease have resulted in an increase in the number of patients with osteoporotic bone tissue causing. amongst other issues, implant loosening. Revision surgery to treat and correct the loosenings should be avoided, because of the additional patient stress and high treatment costs. Shape memory alloys (SMA) can help to increase the anchorage stability of implants due to their superelastic behavior. The present study investigates the potential of hybridizing NiTi SMA sheets with additively manufactured Ti6Al4V anchoring structures using laser powder bed fusion (LPBF) technology to functionalize a pedicle screw. Different scanning strategies are evaluated, aiming for minimized warpage of the NiTi SMA sheet. For biomechanical tests, functional samples were manufactured. A good connection between the additively manufactured Ti6Al4V anchoring structures and NiTi SMA substrate could be observed though crack formation occurring at the transition area between the two materials. These cracks do not propagate during biomechanical testing, nor do they lead to flaking structures. In summary, the hybrid manufacturing of a NiTi SMA substrate with additively manufactured Ti6Al4V structures is suitable for medical implants.

## 1. Introduction

The current research focus on the development of screw implants is a combination of established manufacturing processes, and materials to enhance their function [1,2,3]. In the course of personalization and demographic change, conventional implants cannot always optimally cover the treatment. The aging population poses a particular challenge: osteoporotic bone tissue makes it difficult to achieve primary stability, but long-term stability in terms of the fatigue strength of implants and prostheses [4,5,6]. In spinal surgery, this leads to an increased loosening of screw implants (pedicle screws) during spinal fusion. The aim of osteoporotic treatment is, therefore, to increase the primary stability and fatigue strength of implants. Our first approaches showed that the use of a superelastic Nickel–Titanium shape memory alloy (NiTi SMA) to anchor elements in titanium (Ti6Al4V) bone implants has a great potential to increase primary stability within the bone [7,8,9]. Ti6Al4V is a widely used standard material for bone implants in medical technology, especially in osteosynthesis [10]. The use of nickel–titanium shape memory alloys (thermal and superelastic alloys) have also been widely researched and are used in human medicine [7,11].

In the structuring of NiTi-SMA elements, the abrasive process of laser treatment is used in the manufacture of medical supplies (e.g., manufacturing of stents) and could be applied to our preliminary work [12,13]. As an alternative to the abrasive process, additive laser powder bed fusion (LPBF) technology with Ti6Al4V on NiTi-SMA was investigated in this study, to provide patterning of the anchoring elements. To structure the NiTi SMA anchoring elements, Ti6Al4V powder was used, which is a well-known and established powder material for medical applications due to its good biocompatibility and processability. With regard to the additive manufacturing of Ti6Al4V powder and the associated high heat input during manufacturing, good biocompatibility has also been demonstrated in our own preliminary work [14].

Two main advantages of additive manufacturing are the high design freedom and economic efficiency for low quantities (favorable for personalized applications) [15,16,17]. Therefore, the application of LPBF technology of NiTi-SMA with TTi6Al4V offers a great potential to functionalize standard titanium bone implants. Our preliminary investigations have already demonstrated the biocompatibility of the compound [18].

The aim of the present study was to investigate the potential of hybridizing supere-lastic NiTi SMA sheets with additively manufactured Ti6Al4V anchoring structures using LPBF and to evaluate them biomechanically, to demonstrate its use in orthopedic surgery; see Figure 1.

## 2. Materials and Methods

### 2.1. LPBF Parameter Screening Preparation

The additive manufacturing process was performed with an M2 Cusing (Concept Laser GmbH, Lichtenfels, Germany)—a conventional laser powder bed fusion machine with a build space of 250 × 250 × 280 mm^3^ and a 400 W CW-diode-pumped fibre laser (1070 nm wavelength). Concept Laser GmbH supplies an associated manufacturing CAD-based slicing algorithm to manufacture the surfaces. The laser spot diameter was set to 100 µm, and layer thickness was set to 25 µm. The necessary protective atmosphere was obtained using argon. For this study, Ti6Al4V Grade 5 powder was used (TLS Technik GmbH & Co. Spezialpulver KG, Bitterfeld, Germany)—a titanium alloy was often used for medical applications. The particle size was between 5 and 66 µm, with 90% of the powder particles having an average diameter of 32.9 µm (verified by dynamic picture analysis using a Camsizer X2, Microtrac Retsch GmbH; see Figure 2). The superelastic NiTi SMA substrates (solid baseplate, 0.3 mm and 0.5 mm sheets) were purchased from Ingpuls GmbH.

Four different scanning strategies are meant to create a pyramid-like anchoring structure, made from Ti6Al4V, which penetrates the trabeculated bone tissue, and thus leads to increased anchorage of the implant.

Contour-CLS represents the conventional CAD-based slicing algorithm of Concept Laser, creating non-editable contour scan vectors (no hatch). The other three variations represent adaptive scanning strategies (Contour-CLI, Cross-CLI, Crosshair-CLI), which manually define every single scan vector (editable) to describe the final geometry. Those adaptive scanning strategies are meant to result in less warpage of the NiTi SMA sheet, due to the optimized distribution of energy input. Contour-CLI scan vectors represent the outer contour of the pyramid-like structure. Cross-CLI scan vectors form a continuous diagonal cross, whereas Crosshair-CLI scan vectors form a diagonal cross, which is left blank in the middle section. A visualization of these scanning strategies can be found in Figure 3.

The parameter screening consists of three steps, as follows, yielding to the optimum set of parameters:Evaluation of the most suitable laser power and scanning speed concerning metallurgic connection between Ti6Al4V and NiTi SMA (substrate: solid NiTi SMA baseplate);Evaluation of the most suitable scanning strategy concerning low warpage-substrate: NiTi SMA sheet (thickness = 0.3 mm);Evaluation of the most suitable combination of laser power, scanning speed (Step 1) and scanning strategy (Step 2) concerning low warpage-substrate: NiTi SMA sheet (thickness = 0.5 mm).

Step 1:

The initial parameter screening on a solid NiTi SMA baseplate examined a total of 64 samples. One sample consists of 49 pyramid-like anchoring structures forming a square of 5 × 5 mm^2^. Those samples vary in laser power *P*, scanning speed *v* and scanning strategy. Figure 4 provides an overview of the initial parameter screening with all varied parameters.

After manufacturing, failed samples (e.g., flaked structures) were first eliminated by visual inspection. Following cutting (Discotom and Accutom, Struers GmbH, Willich, Germany), embedding (EpoFix, Struers GmbH, Willich, Germany), grounding and polishing (Tegramin 30, Struers GmbH), the remaining samples were analyzed concerning fusion depth *t_f_* between substrate and structure using light microscopy (Eclipse Me 600, Nikon Instruments, Tokyo, Japan). The evaluation of the most suitable laser power *P* and scanning speed *v* was based on the metallurgic connection between Ti6Al4V structures and NiTi SMA substrate, as well as the respective crack development.

Step 2:

In the second step, the most suitable scanning strategy was evaluated by fabricating four samples with one scanning strategy each and the best respective set of laser parameters (laser power *P* and scanning speed *v*, based on the results of Step 1). A 0.3-mm-thick NiTi SMA sheet served as substrate in order to show significant differences in warping effects, due to the various induced thermal loads. During the manufacturing process, a fixture held down the metal sheet to ensure manufacturability. After removal of the (warped) sample from the machine, an optical 3D scanner (Atos Scan Core, GOM GmbH) measured warping. The maximum warpage was determined via the best-fit comparison of the CAD model and the hybridized NiTi SMA sheet. The evaluation of the most suitable scanning strategy was based on the lowest value of distance *s* between ground and the highest point of the Ti6Al4V structure (see Figure 5), provided no other defects occurred.

Step 3:

Finally, the approach was transferred to the target thickness of the superelastic NiTi SMA sheet (0.5 mm) to fabricate a fully functional sample, ready for biocompatibility testing. The best two of the four previously examined sets of parameters were used for this purpose. Warpage was again measured utilizing optical 3D scanning (Atos Scan Core, GOM GmbH, Braunschweig, Germany). Furthermore, SEM gave an insight into crack development and powder adhesion (Supra 25, Zeiss AG, Jena, Germany). Low warpage, few (or no) cracks between structure and substrate, and low powder adhesion were the decisive criteria for the final evaluation of the best set of parameters.

### 2.2. Biomechanical Analysis Preparation

Before testing, the final functional sample was heat-treated at 500 °C for 30 min to shape the samples and reduce residual stresses during sheet production. Guide values for the shape memory can be taken from the literature. Since the transformation temperature of the superelastic effect increases with increasing duration of the heat treatment, the duration must be kept as short as possible [19]. At the same time, however, the component should be heated through to the interior. High temperatures above 400 °C lead to a decrease in the tensile strength of the component [20], but lower temperatures are not sufficient to completely relieve internal stresses [21]. Therefore, a temperature of 500 °C was established as a compromise between high tensile strength and good mold embossing [22,23]. For the Ti6Al4V, this is also a compromise temperature, which is at the lower limit of the stress-relieving temperature of Ti6Al4V [24,25,26]. After the shape stamping process, the component is quenched in a water bath, since cooling in air can lead to a slight increase in the transformation temperature [27]. Subsequently, the formed structural sheet was cut into 10-mm-wide strips utilizing wire electrical discharge machining (EDM) technology (MV1200S, Mitsubishi Electric, Tokyo, Japan).

To investigate the adhesion of the Ti6Al4V structures to the NiTi SMA sheet under maximum load (deformation), a servo-mechanic tension-compression-torsion machine (servo-mechanic design 10 kN/200 Nm, DYNA-MESS Prüfsysteme GmbH, Stolberg, Germany) was upgraded with an in-house-manufactured 3-point bending test rig. For this, the smallest measuring range of the testing machine was used (0–1000 N), and the focus was placed on displacement measurement. The radii of the supports, as well as the radius of the compression fin (D), are 5 mm. The sample was placed centrally on the sample grips so that the structured surface was oriented downwards; see Figure 6. Following DIN EN ISO 178 and 7438, the superelastic sheets were loaded 20 times, at a rate of 0.166 mm/s with a travel range (s) of 6.6 mm and a bearing distance (L) of 30 mm. The unloading rate was set to 0.25 mm/s. Three identical NiTi SMA samples were examined.

The same test setup was used for a dynamic investigation. The travel range (s) of 2 mm was run a total of 20,000 times. The same test setup was used for a dynamic investigation.

This is used to evaluate the adhesion of the Ti6Al4V structures, and as evidence of the superelastic properties of the NiTi SMA samples after laser patterning (resetting the samples to initial position after loading). For evaluation, the respective loaded sample was examined and compared with an unloaded sample using an optical microscope (Eclipse ME600, Nikon, Tokyo, Japan) with respect to its interface between NiTi SMA sheet and Ti6Al4V anchoring structure. Both samples were functionalized with the same scanning strategy and parameters.

The possibility of abrasion of the Ti6Al4V structures during biomechanical use was examined by means of a screw pullout-test according to “ASTM F543 Specification and Test methods for Metallic Medical Bone Screws”. For this purpose, the hybridized NiTi SMA strips were inserted into specially machined pedicle screws, which were screwed in an artificial bone (Block 10 PCF Cellular, SawBones, Malmö, Sweden). Two of the functionalized pedicle screws were tested three times in each case, in a 4-span jaw of a servo-hydraulic tension-compression-torsion machine.

The pedicle screws were then inserted into the artificial bone at an insertion speed of 18 °/s, with a feed rate of 2.0 mm/s and a final depth of 28 mm; see Figure 7a. The following activation of the integrated NiTi SMA strips (anchoring elements) results in a diameter increase of 1 mm; see Figure 7b. The performance of the final pullout-test took place at a speed of 0.083 mm/s and a path of 28 mm to ensure complete removal of the screw from the artificial bone; see Figure 7c. The test setup is based on our preliminary studies [7,28].

## 3. Results

### 3.1. Results of the LPBF Parameter Screening

Step 1:

The 64 samples were successfully fabricated. However, the cutting process resulted in the exclusion of 29 samples due to flaking (separation of the Ti6Al4V structures from the NiTi SMA baseplate). The microscopic examination of the remaining 35 samples showed a broad range of connection qualities: from insufficient connections (separation of the structures) due to too-low energy input, or overmelting (no formation of pyramid structures) due to too-high energy input. In addition, some specimens showed severe cracking, which also led to exclusion. Figure 8 shows the four best-connected structures for each scanning strategy. Furthermore, some samples showed severe cracking, which also resulted in exclusion. Figure 8 shows the four best-connected structures for each scanning strategy.

Table 1 gives an overview of the laser power and scanning speed parameters for each scanning strategy, which resulted in the best metallurgic connection. Furthermore, the average fusion depth is given.

Step 2:

The parameters of Step 1 (see Table 1) form the basis of additive manufacturing on 0.3 mm thick NiTi SMA sheets. All four samples were successfully fabricated, retaining their superelastic behavior. The measured distances *s* for warpage evaluation ranged between 4.42 mm (Crosshair-CLI) and 6.02 mm (Cross-CLI), as seen in Table 2. Despite the low distance *s* of the Crosshair-CLI sample, it showed cracks between the Ti6Al4V structures and NiTi SMA sheet and was, therefore, excluded from further investigation. The Contour-CLS sample showed severe deformation on the bottom side of the NiTi SMA sheet, and hence was also excluded. This led to the decision to choose Contour-CLI and Cross-CLI for further investigations.

Step 3:

In Step 3, the best two samples were fabricated on a 0.5 mm NiTi SMA sheet with their respective parameters:Contour-CLI; 130 W; 1000 mm/s;Cross-CLI; 160 W; 1000 mm/s.

The fabrication was successfully conducted. Figure 9 depicts the optical 3D measurements of distance *s*. It could be seen that the Contour-CLI sample showed a lower warpage (2.17 mm) than the Cross-CLI sample (2.42 mm). Eventually, the Contour-CLI sample was chosen for further processing and biomechanical analysis (see Section 3.2).

### 3.2. Results of the Biomechanical Analysis

The crack examination of the interface between the NiTi-SMA and Ti6Al4V structure of the static bending test showed that individual material occlusions and cracks between the two materials form during manufacturing. However, after the static maximum bending of the two strips (unloaded strip and loaded strip), no further cracks or crack extensions could be observed (see Figure 10). The crack length remained the same.

The light microscopic examinations showed that cracks had already formed on the Ti6Al4V pyramid-shaped base surface in the unloaded state (before testing) (see Figure 11). After testing, crack elongation of about 50 µm was measured, which is an increase of about 33%.

The pullout-test showed that the additively fabricated Ti6Al4V structures could withstand the hybridized NiTi SMA sheets being placed in an artificial bone under tensile load; see Figure 12.

## 4. Discussion

Implants are exposed to large loads and moments in the body [29,30] in order to sufficiently support the biomechanical system. Especially in spinal surgery, large loads occur on the implant and generate micro-movements between the implant and bone tissue. These micro-movements can vary in magnitude depending on the load and must be sustained by the implant. In the context of a hybrid connection within the implant, it is, therefore, important to examine whether this connection is of sufficient quality to avoid e.g., flaking of structures. Foreign bodies can trigger inflammatory reactions and damage the surrounding tissue, even leading to screw (implant) loosening. Our own clinical observations showed that, in osteoporotic bone, these foreign bodies can diffuse further into the bloodstream into the heart, where cardiac arrest can occur. The static 3-point bending test was able to simulate the maximum load (deflection) of the hybridized sheet. It could therefore be demonstrated that the Ti6Al4V structures do not flake and no additional cracks are formed at a maximum deflection of the superelastic NiTi SMA sheet of 6%. The investigation serves as a safety verification, because a deflection of 6.66 mm will not be feasible in the application within a pedicle screw. This is due to the high compression of the surrounding bone tissue as well as the available space in the pedicle.

In addition to the investigation of the maximum load, it was important to show that the connection also lasts under cyclic loading (dynamic testing). Spinal implants usually remain in the body for a long time and will be exposed to daily loads. Days or weeks can pass before stable ingrowth or regrowth of bone tissue occurs. During this time, the micro-movements between bone tissue and implant are particularly large, so that increased stress on the structures and, therefore, loosening of the implant or parts of the implant, respectively, can occur. The study simulates the first two weeks after implantation with 20.000 test cycles. This corresponds to about 1430 steps per day. Due to the low postoperative movement and impairment of the patients, approximately one third of the movement (on average, 5200 steps/day are covered in Germany [31], was calculated. The deflection was set as a path of 2 mm, since the implant is to be examined under a “normal condition” (maximum bulge of 1 mm required). This setup includes a safety factor of two. It could be proven that the structures can withstand a permanent load. However, the crack formation increases. This can be relativized by the fact that, under physiological conditions, the deflections do not occur to such an extent.

The final application scenario was simulated with synthetic bone and a functionalized pedicle screw. The aim was to demonstrate that the structures in a bone-like tissue hold to the plate after extraction in an erect form (worst-case scenario). The simulated test process is a standard test of screw implants and must be successfully passed for the approval of new implants. It provides a quick and application-oriented statement as to whether the structures can hold under (bio-)mechanical parameters, surrounded by artificial bone tissue. In contrast to the standard test procedure, with only one required repetition, the present study conducted three repetitions. Accordingly, it was demonstrated that the structures could withstand multiple cycles without flaking, and could thus be applied for use in the body.

The mechanical tests showed good to very good durability of the structures on the sheet regarding microscopic screenings before and after testing. Further tests, such as a toggle test and the use of human specimens, are planned and must be carried out for conclusive statements. The hybrid combination of biocompatible, medical standard materials (Ti6Al4V powder and NiTi SMA) using LPBF showed that it is mechanically suitable to manufacture anchoring elements made of Ti6Al4V on NiTi SMA. In addition, our own preliminary investigations proved that the resulting structure, made of NiTi and Ti6Al4V, is biocompatible [18].

## 5. Conclusions

Hybrid production of NiTi SMA substrate and additively manufactured Ti6Al4V structures can be applied to medical implants. The optical and biomechanical investigations showed a good connection between Ti6Al4V anchoring structures and the NiTi SMA sheet. This was demonstrated on a functionalized pedicle screw. Our preliminary studies also showed that the compound is biocompatible and suitable for use in the human body. Differential scanning calorimetry (DSC) measurements must be performed in further studies to investigate the transformation temperatures before and after laser treatment. The tests performed in this investigation are only suitable for the mechanical and superelastic characterization of the hybrid Ti6Al4V-NiTi SMA connection. For further optimization, e.g., in terms of laser power and mechanical properties, DSC measurements have to be investigated. In addition, further biomechanical investigations, such as toggle and fatigue tests (according to ASTM 1717-15 and ASTM 543), should also be carried out in order to obtain further information regarding the fatigue strength of the connection. Furthermore, long-term tests are useful, for example, to determine the corrosion properties.

Continual investigations should include an analysis of remaining powder adhesion, further optimization of additive manufacturing process parameters (laser power, scanning speed) and scanning strategies, and geometrical modifications and the introduction of heat treatment to reduce residual stresses and notch effects. Another approach could be the selection of the same material, i.e., NiTi SMA powder (anchoring structures) and NiTi SMA substrate (sheet/strip). This would likely result in even larger possible bending radii and allow for further applications, such as applications in hip arthroplasty (new biocompatibility studies are needed).

## Figures and Tables

**Figure 1 materials-14-03098-f001:**
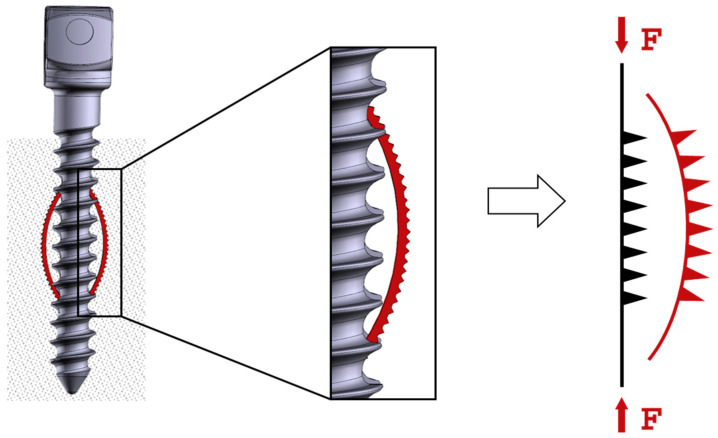
Operating principle of the hybridized anchoring element [18].

**Figure 2 materials-14-03098-f002:**
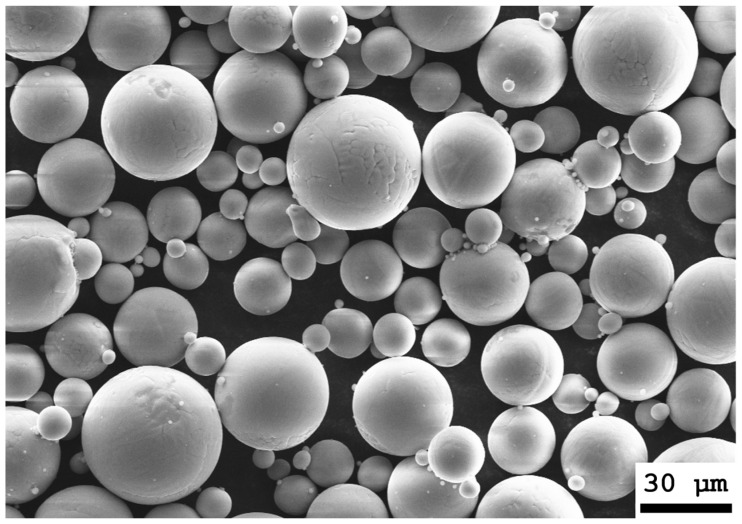
SEM image (Supra 25, Zeiss AG) of the processed Ti6Al4V Grade 5 powder.

**Figure 3 materials-14-03098-f003:**
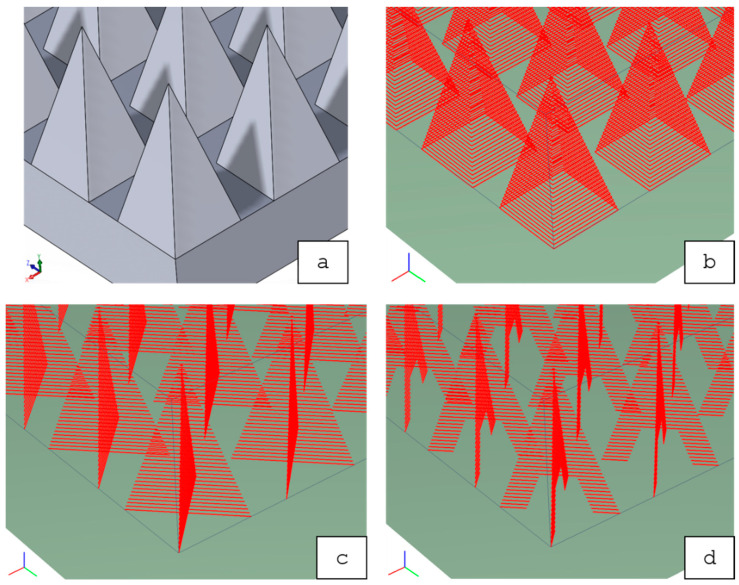
Overview of the four scanning strategies (red lines = scan vectors): (**a**) Contour-CLS (scan vectors [not shown in the sub-figure] are created automatically based on the shown CAD-data); (**b**) Contour-CLI; (**c**) Cross-CLI; (**d**) Crosshair-CLI.

**Figure 4 materials-14-03098-f004:**
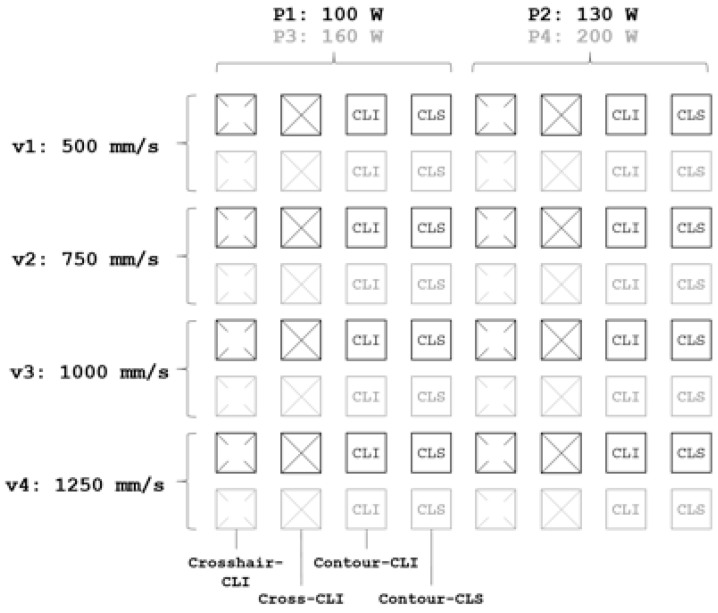
Overview of the initial parameter screening setup (Step 1).

**Figure 5 materials-14-03098-f005:**
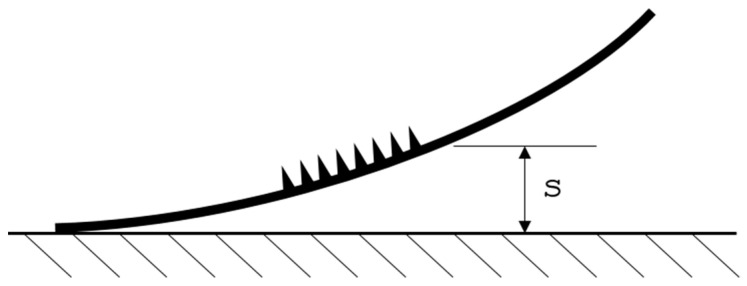
Measurement method to evaluate distance *s*.

**Figure 6 materials-14-03098-f006:**
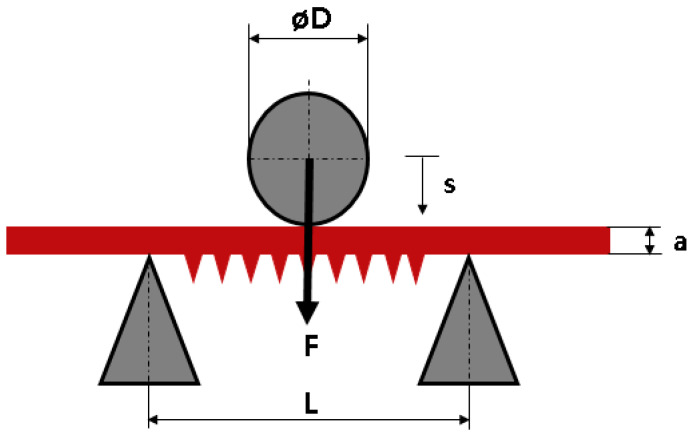
Schematic 3-point bending test-setup; D: diameter of the bending die; L: bearing distance; a: sheet thickness; s: travel range of the bending die.

**Figure 7 materials-14-03098-f007:**
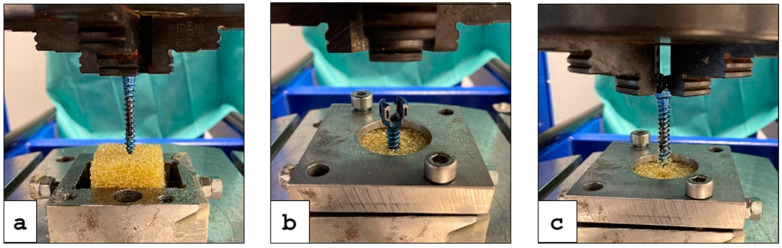
Test procedure: (**a**) Insertion of the functionalized pedicle screw in the artificial bone (yellow block); (**b**) Activation of the NiTi SMA strips (anchoring elements); (**c**) Pullout-test of the screw.

**Figure 8 materials-14-03098-f008:**
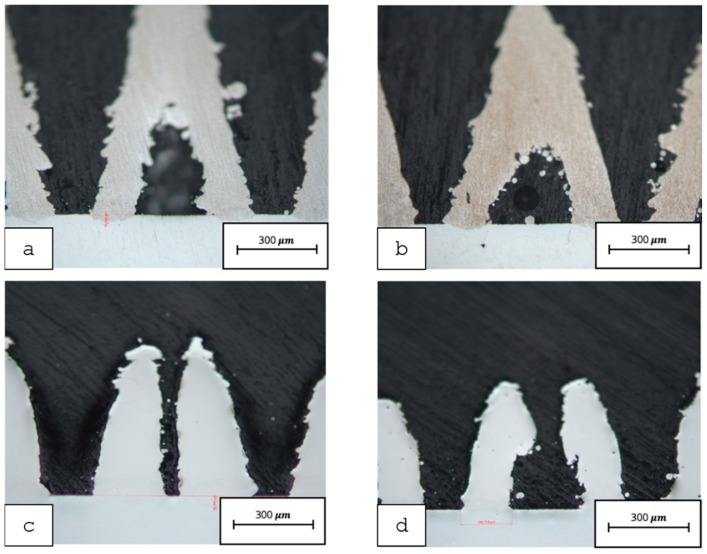
Light microscopy images of all four scanning strategies in the context of the initial parameter screening (solid NiTi SMA baseplate): (**a**) Contour-CLS; (**b**) Contour-CLI; (**c**) Cross-CLI; (**d**) Crosshair-CLI.

**Figure 9 materials-14-03098-f009:**
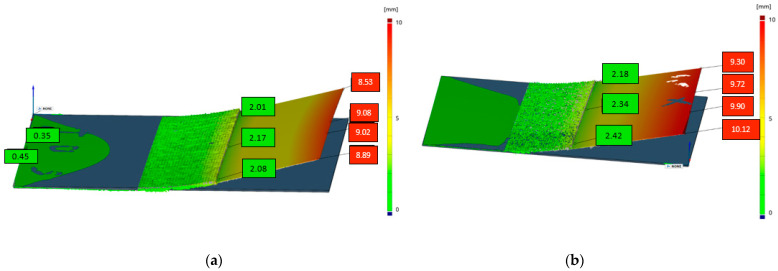
Optical 3D scanning images for warpage measurement (distance *s* [mm]): (**a**) The scanning strategy Contour-CLI showed lower maximum warpage with 2.17 mm than (**b**) Cross-CLI with 2.42 mm.

**Figure 10 materials-14-03098-f010:**
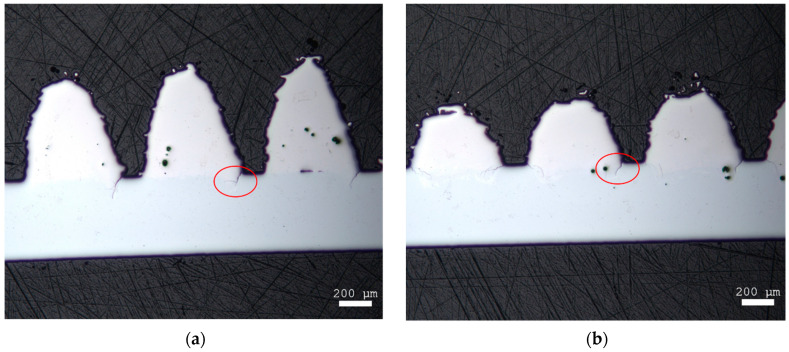
Microscope images of the hybridized NiTi SMA strips (**a**) before and (**b**) after static testing.

**Figure 11 materials-14-03098-f011:**
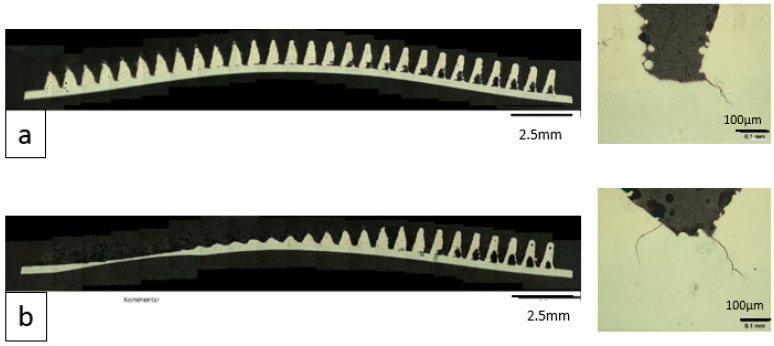
Microscope images of the hybridized NiTi SMA strips before and after dynamic testing: (**a**) Before: overview image (left); detailed image (right); (**b**) After: overview image (left); detailed image (right).

**Figure 12 materials-14-03098-f012:**
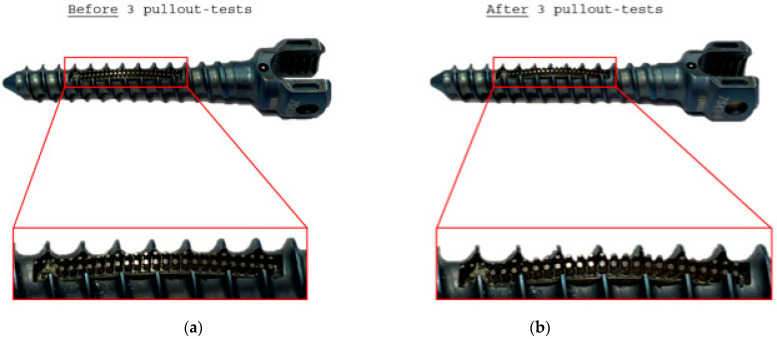
Detailed images of the functionalized pedicle screw with integrated hybridized NiTi SMA sheets: (**a)** Before and (**b**) after three pullout tests.

**Table 1 materials-14-03098-t001:** Selected optimum parameters for laser power *P* and scanning speed *v* for each scanning strategy and the respective average metallurgic fusion depth *t_f_*.

	Contour-CLS	Contour-CLI	Cross-CLI	Crosshair-CLI
*P* [W]	100	130	160	160
*v* [mm/s]	500	1000	1000	1250
*t_f_* [µm]	35.0	33.3	44.9	35.4

**Table 2 materials-14-03098-t002:** Measured distances *s* representing the degree of warpage for each scanning strategy (substrate: 0.3 mm NiTi SMA sheet).

	Contour-CLS	Contour-CLI	Cross-CLI	Crosshair-CLI
*s* [mm]	4.61 ^1^	4.61	6.02	4.42 ^2^

^1^ Severe deformation on bottom side of the NiTi SMA sheet → excluded. ^2^ Poor connection between Ti6Al4V structures and NiTi SMA sheet → excluded.

## Data Availability

Link to the to publicly archived datasets: Investigation into the Hybrid Production of A Superelastic Shape Memory Alloy with Additively Manufactured Structures for Medical Implants.

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
