# Peer review of "Investigation into the Hybrid Production of a Superelastic Shape Memory Alloy with Additively Manufactured Structures for Medical Implants"

_materials, 2021, doi:10.3390/ma14113098_

Round 1
Reviewer 1 Report
The paper presented a hybrid production of super elastic shape memory alloy (SMA) with additively manufactured structures for medical implants. The introduction is written clearly with adequate support of literature. The story line flows naturally and the paper reads really well. The experimental section is clearly presented and the reviewer was able to understand the paper without any difficulty. The steps for selecting the optimum parameters of laser powder bed fusion (LPBF) technology is clearly presented. Biometrical parameters were investigated including adhesion test, 3 point bending, pull out test and SEM image analysis. It was concluded that hybridization of NiTi SMA with Ti6Al4V using LPBF is mechanically suitable for the fabrication of Ti6Al4V anchoring elements on NiTi SMA. The conclusion was drawn based on experimental observation. The paper will definitely attract citation and it was written clearly and concise English. Few minor comments:
- Please avoid excessive grouped citation [1–7], remove at least 3 of them or write one sentence per references by giving them proper credit.
- There is a mixture of figure and fig. in the manuscript. Please use only one form.
- Please remove any references from conclusion.
Author Response
Dear Reviewer #1
Thank you for your review.
We have modified the paper as follows (28.06.2021):
- We have reduced the grouped quotes and thinned them out in terms of priority.
- We have standardized the labels and references for the figures.
- We have removed the source citations from the conclusion.
In addition, we have expanded the introduction in relation to other reviewer comments and explained the process, stress relief annealing and mold embossing.
The modification in the manuscript have been inserted using "track changes" in Word. We hope the revised manuscript is suitable for publication.
Isabell Hamann
Reviewer 2 Report
Dear authors,
This paper is well written and I would recommend this paper to be published on this journal. This paper succeeds in introducing the availability of a combination of NiTiSMA substrate with Ti6Al4V structures through AM technologies for medical implants. I just have two small points to claim:
(1) Line 158: why this temperature, 500oC, and this time, 30min, were chosen by you for heat treatment? How about the cooling process? I would advise you to add some explanation here If possible. Have these parameters in your previous research been demonstrated to reduce the residual stress.
(2)Line 332: bevor->before.
Author Response
Dear Reviewer #2
Thank you for your review.
We have modified the paper as follows (28.06.2021):
Starting from line 172, we have explained the process of mold embossing and reduction of residual stress by heat treatment and added content.
A temperature of 500°C was chosen for the shape forming and simultaneous stress reduction, as this is a good compromise value for the NiTi but also for Ti6Al4V to reduce residual stresses and at the same time allow shape forming. The time period of 30 min was chosen to allow uniform heating through of the specimen.
The cooling process takes place in a water bath of 20°C, since cooling in air can lead to a slight increase in the transformation temperature (cooling too slowly).
For a more understandable overview and content of the paper, we have additionally expanded the introduction.
The modification in the manuscript have been inserted using "track changes" in Word. We hope the revised manuscript is suitable for publication.
Isabell Hamann
Reviewer 3 Report
Manuscript ID: materials-1248190
Title:
Investigation into the hybrid production of a superelastic shape memory alloy with additively manufactured structures for medical implants
Authors:
Isabell Hamann * , Felix Gebhardt , Manuel Eisenhut , Peter Koch , Juliane Thielsch , Christian Rotsch, Welf-Guntram Drossel , Christoph-Eckhard Heyde, Mario Leimert
Review Report
The introduction is very short. An uninterested reader does not have a sufficient overview of the mentioned topic.
It is necessary to characterize the materials used in more detail, to describe their properties. It would be appropriate to add data on other similar materials.
Next, describe the methods by which the other properties of these substances can be determined. The other mechanical properties of materials can be tested (tensile strength, compressive strength, flexural strength, Young's modulus, etc.). And last but not least, the study of corrosion properties (long-term corrosion, electrochemical corrosion, ...) is very important.
I have no comments on the experimental part.
Author Response
Dear Reviewer #3
Thank you for your review.
We have modified the paper as follows (31.05.2021):
For a better understanding and overview of this paper, we have extended the introduction. In this regard, we have also commented again on the materials used and their mechanical properties. In this study, only biocompatible, tested and recognized standard materials of medical technology were used. Using the LPBF process, these materials (Ti6Al4V and NiTi SMA) were connected to each other. The process for additive manufacturing of Ti6Al4V for medical implants is already state of the art and biocompatible. Therefore, the focus in this study was to investigate the (bio)mechanical connection of the two materials using additive manufacturing. In a previous study, biocompatibility had already been demonstrated and cellular studies investigated, so that here only the functional/ mechanical strength was investigated under biomechanical aspects.
Supplementary investigations, such as endurance, toggle or long-term tests to evaluate the corrosion properties would, of course, have to be carried out in further studies. werden.
The modification in the manuscript have been inserted using "track changes" in Word. We hope the revised manuscript is suitable for publication.
Isabell Hamann